# Analysis of Zinc and Copper Content in Selected Tissues and Organs of Wild Mallard Ducks (*Anas platyrhynchos* L.) in Poland

**DOI:** 10.3390/ani14081176

**Published:** 2024-04-14

**Authors:** Elżbieta Bombik, Antoni Bombik, Katarzyna Pietrzkiewicz

**Affiliations:** Faculty of Agricultural Sciences, University in Siedlce, Prusa Street 14, 08-110 Siedlce, Poland; antoni.bombik@uws.edu.pl (A.B.); pietrzkiewicz.ka@gmail.com (K.P.)

**Keywords:** mallard duck, muscles, liver, zinc, copper, bioindicator of environmental pollution

## Abstract

**Simple Summary:**

Analysis of zinc and copper levels in the tissues of wild animals, including mallard ducks, is important for assessing the degree of contamination of the environment with these elements. This problem is worth addressing with regard to the safety of consumers of game meat. It was concluded from the study that the Leszno hunting district is more polluted than the Siedlce hunting district. The significantly higher average zinc and copper concentrations in the liver of mallards harvested in this hunting district may have been influenced by fertilization of crop fields with fertilizers containing these elements and by extraction of ore containing zinc and copper minerals. Statistically significant differences were also shown in the average level of copper in the leg muscles of males compared to females.

**Abstract:**

The research material included selected muscles and liver of mallard ducks obtained in two research areas. A total of 28 mallards were obtained for the study—six males and six females from the Siedlce hunting district and eight males and eight females from the Leszno hunting district. Zinc and copper concentrations were determined by inductively coupled plasma optical emission spectrometry (ICP OES). It was concluded from the study that the Leszno hunting district is more polluted than the Siedlce hunting district. Among the examined tissues of mallard ducks from both hunting districts, the highest content of zinc and copper was found in the liver. In this organ, birds shot in the Leszno hunting district were characterized by a significantly higher content of these elements compared to birds shot in the Siedlce hunting area. The significantly higher average zinc and copper concentrations in the liver of mallards harvested in this hunting district may have been influenced by fertilization of crop fields in this area with fertilizers containing these elements and by extraction of ore containing zinc and copper minerals in the Legnica-Głogów Copper District. This is an important problem with regard to the safety of consumers of game meat. It was shown that the tissues of male mallards were characterized by higher average levels of zinc and copper than those of females, but the differences were statistically significant only in the case of the average copper content in the leg muscles.

## 1. Introduction

The concentration of zinc (Zn, Latin zincum) in uncontaminated soils of Poland ranges from 50 to 100 mg·kg^−1^ [1]. Its concentration in soils increases as a result of industrial and agricultural activity. The average zinc level in soil is about 64 mg·kg^−1^ and varies widely depending on the type of soil [2]. Grzebisz et al. [3] showed that the average zinc content in the urban soils of Poznań was 72.9 mg·kg^−1^, which was 61% higher than the biogeochemical background value for this element (36.7 mg·kg^−1^). According to Kabata-Pendias [4], the most contaminated soils in Poland are found in industrial areas producing non-ferrous metals, where zinc concentrations reach 13,800 mg·kg^−1^, and areas with zinc ore mines, where the zinc level in the soil is up to 3200 mg·kg^−1^.

According to Kabata-Pendias and Pendias [1], the average zinc concentration in potable groundwater in Poland is estimated at 15 µg/L, with considerable levels noted in river bottom sediment. In the Warsaw metropolitan area, the zinc concentration in the sediments of the Vistula reached 6500 mg·kg^−1^.

Zinc from the soil enters the bodies of animals mainly via plants. It is ingested from water directly or with phyto- and zooplankton. For plants, it is more easily available from anthropogenic sources than from natural ones. The largest amounts of this element are found in the roots of plants. The average zinc level in grasses ranges from 25 to 47 mg·kg^−1^, e.g., from 24 to 39 mg·kg^−1^ in clover and from 18 to 33 mg·kg^−1^ in cereal grains [2].

Zinc is a ubiquitous element in the body. It is present in all organs, tissues, skin appendages, body fluids and semen. It plays a key role in many enzymatic reactions, acting as an activator of various enzymes or participating in the composition of enzymes [5]. It is associated with intracellular processes. It is involved in biosynthesis of proteins, DNA and RNA and in the metabolism of nucleic acids. As a component of nearly 200 proteins and metalloenzymes (including carbonic anhydrase, which is essential for Ca metabolism), it is an important activator of metabolism [6]. It is responsible for perception of flavour and odour. In addition, it stimulates the immune system, exhibits antioxidant activity, and influences reproduction [7,8]. During its bioaccumulation, zinc competes for the same transporters as cadmium and lead, which means that it acts as their antagonist and performs an important protective role in the case of poisoning with these heavy metals [9]. In addition, soil fertilization with high levels of phosphorus and liming reduce the mobility of zinc in the soil and can cause secondary zinc deficiency in plants [1].

Zinc deficiency (hypozincaemia) in both people and animals leads to disturbances of biosynthesis of proteins and nucleic acids, cellular metabolism disorders, reduced antibody production, poor wound healing, disturbances of keratinization of skin appendages, decreased libido, and inhibition of the growth of young organisms [10]. Zn deficiency negatively affects reproduction and causes disturbances in embryonic development. Poorer feed conversion can be observed in animals, and a decrease in laying and hatching rates and disturbances in embryonic development have been noted in chickens [11]. Zn is an element with low toxicity for the body. Zn concentrations above 100 ppm in the body result in disease symptoms [12]. Excessive zinc causes nausea, vomiting, weight loss, weakness, and anaemia [8,9], and can also lead to death [9]. The zinc level in animals depends on factors such as the type of muscle or tissue, the animal’s breed and genotype, diet, and the zinc concentration in the environment [13,14,15,16,17]. Copper (Cu, Latin cuprum) is a dark gold chemical element belonging to the potentially toxic elements. [18]. It is counted among both heavy metals and essential microelements for the functioning of living organisms. Its average content in the Earth’s crust is estimated at 55 mg/kg. Copper in the natural environment is present mainly in the form of minerals [2]. The concentration of this element in soils of Poland ranges from 0.2 to 725.0 mg·kg^−1^ [19].

The copper level in the bottom sediments of polluted water bodies and rivers can reach 2000 mg·kg^−1^, while in unpolluted waters it does not exceed 100 mg·kg^−1^ [4]. This metal is easily accumulated in phyto- and zooplankton. According to Khellaf and Zerdaoui [20], gibbous duckweed (*Lemna gibba* L.) can remove up to 80% of Cu from water.

Copper in the body is bound to proteins and pigments in metalloenzymes. It is found in large quantities in the bones, muscles, skin, skin appendages, and liver. The liver is the largest storehouse of this element, with nearly 10% of the entire pool of Cu ions in the body [21]. Copper is a component and activator of numerous enzymes, including ceruloplasmin, superoxide dismutase (SOD), hydroxylase, cytochrome and lysine oxidase, and tyrosinase [22]. Cu also takes part in haematopoiesis, synthesis of collagen, keratin, melanin and neurotransmitters, and mitochondrial respiration [22,23,24]. Cu absorption can be disturbed by the presence of antagonistic elements such as zinc, sulphur, molybdenum, iron and calcium [2,9].

Cu in the body, with the help of low-molecular-weight cytoplasmic proteins (metalloproteins), is accumulated mainly in the liver and kidneys [25]. Its level in the peripheral blood increases due to liver damage. Symptoms of copper deficiency in people and animals include anaemia, skin and hair depigmentation, diarrhoea, muscle weakness, limited growth, cardiovascular disease, nervous system abnormalities, and reproductive disorders [21,26]. Copper is counted among conditionally toxic elements. In the diet of animals, it has been called a ‘cumulative poison’. Excessive copper in the body induces a number of negative effects, including diarrhoea, nausea, liver and kidney damage, and neurological changes [21].

The intake of zinc from the adult diet in European countries ranges from 8.0 to 14.0 mg/day [27], while copper intake ranges from 1.15 to 2.07 mg/day [28]. It was assumed that the efficiency of zinc absorption from mixed diets is 30%. Based on balance studies, it was concluded that the absorption of copper from the diet is approximately 50% for all age groups of people [29].

The accumulation of heavy metals in the body of wild birds depends on the environmental conditions in which these species live, the type of diet, as well as different exposure to environmental pollution from habitats surrounded by pollutant emitters (transport, industry and agriculture) [30].

Wild mallard duck meat is very tasty, lean, has lower fat content, a favourable fatty acid profile and a higher protein content than farmed animals [31,32]. In Poland, 45,235 wild ducks were shot in the 2021/2022 hunting season [33]. The per capita consumption of this meat is not high. It is not available on the market, and it is a niche product. In Poland, wild mallard duck meat is eaten mainly by hunters and their companions [32]. The available literature lacks well-documented sources regarding the relationship between the content of zinc and copper in the tissues of mallard ducks with the place of harvest and the sex of these birds.

The aim of the study was to analyse the content of zinc and copper in selected tissues and organs of mallard ducks depending on the sex and place the birds were obtained. It was assumed that the mallard duck may be a good bioindicator of environmental pollution.

## 2. Materials and Methods

### 2.1. Animals and Sample Collection

The material for the study included the breast and leg muscles and the liver of mallard ducks shot in two research areas—the Siedlce and Leszno hunting districts. Both regions were distinguished by different environmental resources and different intensification of agriculture. The Greater Poland Voivodeship is characterized by large farms conducting intensive plant production, including a high level of mineral fertilizers. Agricultural farms in the Masovian Voivodeship are smaller and managed in a more sustainable way, which means that their impact on the environment is lower [32]. The shooting of mallards was limited to the first two months of the hunting season for this species, i.e., from 15 August to 15 October 2017, before the migration of these birds began. The age of the ducks was assessed by the degree of hardness of the beak. Most of them were young ducks (about one year) because they had more plastic beaks. A total of 28 mallards were obtained for the study: six males and six females from the Siedlce hunting district and eight males and eight females from the Leszno hunting district. Frozen samples were stored at −20 °C. The research material was collected in accordance with the requirements of the National Ethics Committee for Animal Experiments of the European Union (authorization nos. 37/2001 and 36/2011).

### 2.2. Laboratory Analysis

Zinc and copper concentrations were determined by inductively coupled plasma optical emission spectrometry (ICP OES) using the Perkin Elmer Optima 2000 DV spectrometer (Shelton, WA, USA). The analytical wavelengths and instrument detection limits: Zn 206.200 nm; DL 10 μg·L^−1^, Cu 324.752 nm; DL 1 μg·L^−1^. The material was thawed and then homogenized in an agate mortar. Weighed samples weighing approximately 1 g were transferred to quartz pressure vessels to which 5.0 mL of 65% HNO_3_ (Suprapur^TM^, Merck KGaA, Darmstadt, Germany) and 1 mL of 30% H_2_O_2_ (Suprapur^TM^, Merck KGaA, Darmstadt, Germany) were added. The closed vessels were placed in a mineralizer controlling temperature and pressure. The cooled and degassed mixture (CO_2_, NO_2_) was made up to 10 mL in class A volumetric flasks (BRAND, Wertheim, Germany). Radiation emission was measured in the prepared solutions. The content of Zn and Cu was determined in mg·kg^−1^ WW (wet weight).

### 2.3. Statistics

Basic descriptive statistics were determined for hunting districts and sex of wild mallard ducks, i.e., arithmetic mean ($\bar{x}$), range of variability (x_min._–x_max._), standard deviation (s) and coefficient of variation (V%). A two-way non-orthogonal analysis of variance (Fisher–Snedecor F test) with interaction according to the mathematical model was performed: y_ijl_ = m + a_i_ + b_j_ + ab_ij_ + e_ijl_,where:

y_ijl_—value of trait for i^th^ hunting district (a = 2), j^th^ sex (b = 2) and lth replicate (measurement)m—population meana_i_, b_j_—main effects of factors, i.e., hunting district and sexab_ij_—effect of interaction of hunting district and sexe_ijl_—sampling error

Significant effects were compared using Tukey’s test for a significance level of 0.05.

## 3. Results

The results of the analysis of zinc in the tissues of mallard ducks (*Anas platyrhynchos* L.) are presented in Table 1 and Table 2.

Analysis of the average zinc level in the breast muscle of wild mallard ducks showed a higher level in males (11.237 mg·kg^−1^ WW) than in females (9.736 mg·kg^−1^ WW), although these differences were statistically non-significant. The average zinc concentration was higher in the breast muscle of mallards shot in the Leszno hunting district (11.055 mg·kg^−1^ WW) than in those from the Siedlce hunting district (9.728 mg·kg^−1^ WW); however, these differences were not statistically significant. The average level of zinc in the pectoral muscle in both hunting districts was higher, although statistically insignificant, in males than in females, and the diversity of this feature was very large in males (50.29%) compared to females (15.18%). Differences in the average zinc level in the leg muscle of mallards between males (28.886 mg·kg^−1^ WW) and females (24.356 mg·kg^−1^ WW) were evident, but statistically non-significant. The average zinc content in leg muscles was significantly higher in mallard ducks shot in the Leszno hunting district (30.605 mg·kg^−1^ WW) than in the Siedlce hunting district (21.309 mg·kg^−1^ WW). The coefficient of variation for the zinc content in the leg muscle of mallards for the hunting districts ranged from 24.63% in the Siedlce hunting district to 29.53% in the Leszno hunting district. In the Siedlce hunting district, the average zinc levels in the leg muscle of female and male mallards were comparable, while in the Leszno hunting district the zinc level was higher—although not significantly—in males than in females. The variation in this trait was similar for the sexes in both hunting districts.

Analysis of the average zinc content in the liver of mallards showed a higher average level in males (35.977 mg·kg^−1^ WW) than in females (34.772 mg·kg^−1^ WW), although these were statistically non-significant differences. The average level of zinc in the liver was significantly higher in birds caught in the Leszno hunting district (39.064 mg·kg^−1^ WW) than in the Siedlce hunting district (30.455 mg·kg^−1^ WW). The coefficient of variation of the zinc level in the liver of mallards for the hunting districts ranged from 19.75% in the Siedlce hunting district to 24.82% in the Leszno hunting district. The average zinc content in the liver of wild mallard ducks in the Siedlce hunting district was higher in males than in females, while in the Leszno hunting district the opposite relationship was found. In both hunting districts, the differences were statistically non-significant. The variation in the content of this element ranged from 13.90% in the Siedlce hunting district in males to 26.05% in the Leszno hunting district in females.

The results of the analysis of copper concentrations in the tissues of mallard ducks (*Anas platyrhynchos* L.) are presented in Table 3 and Table 4.

A higher, although statistically insignificant, average concentration of copper in the breast muscle of mallards was found in males (4.706 mg·kg^−1^ WW) than in females (4.537 mg·kg^−1^ WW). The average copper level in this tissue was significantly higher in mallards shot in the Leszno hunting district (5.285 mg·kg^−1^ WW) than in those obtained in the Siedlce hunting district (3.738 mg·kg^−1^ WW). The coefficient of variation of copper content in the breast muscle of mallards for the hunting districts ranged from 26.34% in the Siedlce hunting district to 28.57% in the Leszno hunting district. In the Siedlce hunting district, the average copper level in the breast muscle was higher in female mallard ducks than in males, while the reverse pattern was noted in the Leszno hunting district. In both hunting districts, these differences were statistically non-significant. The variation in this trait was high, especially in the Siedlce hunting district—from 14.62% in females to 34.57% in males.

Analysis of the average copper content in the leg muscle revealed a significantly higher level in males (2.637 mg·kg^−1^ WW) than in females (2.074 mg·kg^−1^ WW). The coefficient of variation of the copper level in the leg muscle of mallards ranged from 16.76% in females to 21.57% in males. Differences in the average copper level in the leg muscle of mallards between individuals obtained in the Siedlce hunting district (2.361 mg·kg^−1^ WW) and the Leszno hunting district (2.351 mg·kg^−1^ WW) were evident, but statistically non-significant. In the Siedlce and Leszno hunting districts, the average copper content in the leg muscles was higher, although statistically insignificant, in male mallards than in females. The variation in the content of this element ranged from 14.51% to 26.78% in males in the Leszno and Siedlce hunting districts, respectively.

In the liver of the mallards, the average copper concentration was higher in males (31.886 mg·kg^−1^ WW) than in females (31.402 mg·kg^−1^ WW), although these differences were statistically non-significant. The average level of copper in the liver was significantly higher in mallard ducks shot in the Leszno hunting district (41.950 mg·kg^−1^ WW)—more than twofold—than in those obtained in the Siedlce hunting district (17.904 mg·kg^−1^ WW). The coefficient of variation of the copper levels in the liver for the hunting districts ranged from 53.72% in the Leszno hunting district to 59.33% in the Siedlce hunting district. In the Siedlce hunting district, the average copper concentration in the liver was higher in male mallards than in females, while the reverse pattern was noted for the Leszno hunting district. In both hunting districts, these differences were statistically non-significant. The variation in the content of this element was high, from nearly 50% to slightly over 60%, in both females and males in both hunting districts.

## 4. Discussion

The present study showed no statistically significant differences between the average zinc concentrations in the breast muscle of mallard ducks shot in the Leszno hunting district and the Siedlce hunting district. Male mallards had higher average levels of this element in the breast muscle than females, although the differences were not statistically significant. Higher average zinc concentrations in the breast muscle of mallards than those obtained for the Siedlce and Leszno hunting districts were observed by [34,35]. Szymczyk and Zalewski [34] showed an average zinc concentration of 19.803 mg·kg^−1^ DW in the breast muscle of mallards obtained in Silesia. Bojar and Bojar [35] obtained the highest average zinc content in the skeletal muscles of mallards harvested in Przytoczno (50.0 mg·kg^−1^ WW). It was more than twice as high as in the muscles of individuals harvested in Mosty (24.6 mg·kg^−1^ WW) and in Częstoborowice (16.0 mg·kg^−1^ WW). This may reflect the higher degree of anthropogenic contamination of the environment with this element. A lower level of zinc (9.11 mg·kg^−1^ WW) was shown in the breast muscle of mallards harvested in Warmia-Masuria by Zalewski et al. [36], by Alipour et al. [15] in mallards harvested in Kanibarazan in Iran (20.6 mg·kg^−1^ DW), and by Dżugan et al. [37] in common pheasants harvested near Szczecin (4.571 mg·kg^−1^ WW). Kokoszyński and Bernacki [13] observed higher average zinc concentrations in the breast muscle of Pekin ducks (0.07 g·kg^−1^ DW) for strain P11 and for strain P22 compared to the present study. The average content of this element in the leg muscle ranged from 140 to 160 mg·kg^−1^ DW, while the breast muscle contained 60 to 80 mg·kg^−1^ DW. Kokoszyński et al. [38] showed average content of this micronutrient in the breast muscle of Pekin ducks ranging from 47.7 mg·kg^−1^ DW for type SM3 Heavy to 74.4 mg·kg^−1^ DW for type Star 53 H.Y. In the present study, mallard ducks acquired in the Leszno hunting district had significantly higher average zinc content in the leg muscle (30.605 mg·kg^−1^ WW) than birds obtained in the Siedlce hunting district (21.309 mg·kg^−1^ WW).

Differences in the average zinc level in the leg muscle of mallards between males and females were considerable but statistically non-significant. Kokoszyński and Bernacki [13] reported higher average zinc content in the leg muscle of Pekin ducks than was found in the present study in mallard ducks of both sexes: 0.14 g kg^−1^ DW for strain P11 and 0.15 g kg^−1^ DW for strain P22. Kokoszyński et al. [38] showed higher average zinc content in the leg muscle of SM3 Heavy (110 mg·kg^−1^ DW), Star 53 H.Y. (125 mg·kg^−1^ DW) and AF51 (106 mg·kg^−1^ DW) Pekin ducks than was observed in mallards in the present study. The higher average zinc content in the leg muscle of Pekin ducks may have been due to the fact that zinc deficiencies are common in plant-based feeds, and zinc bioavailability may be limited by high fat content. For this reason, this element is added to feed in the form of easily digestible salts.

The average zinc level in the liver of mallard ducks harvested in the Leszno hunting district was also significantly higher than in birds from the Siedlce hunting district. This may have been due to higher levels of application of phosphate fertilizers in the Greater Poland Voivodeship, which in the 2015/16 marketing year amounted to 44,900 t, in comparison to 38,420 t in the Masovian Voivodeship [39,40]. According to [2], the amount of zinc introduced to the soil with phosphate fertilizers ranges from 50 to 1450 mg Zn/kg. A higher concentration of this element occurs in the tissues and organs of mallard ducks obtained in the Leszno hunting district and may also have been influenced by extraction of ore containing zinc minerals in the Legnica-Głogów Copper District, located about 80 km away.

The liver of mallards had higher average zinc content than the breast and leg muscles, because zinc in vertebrates is initially accumulated in the liver, after which it is deposited in the kidneys and sex glands. Differences in the average zinc concentration in the liver between male and female mallards were statistically non-significant, which is in agreement with findings reported by [17] in individuals of this species harvested in autumn in eastern Austria. Sujak et al. [41], in the liver of mallard ducks harvested near Lubartów, noted an average zinc concentration of 113.53 mg·kg^−1^ DW.

Lower average zinc concentrations in the liver of mallards than in the present study were obtained by [34] in birds harvested in Warmia-Masuria and in Silesia (24.288 mg·kg^−1^ DW and 26.112 mg·kg^−1^ DW, respectively), by [36] in mallards in Warmia-Masuria (21.49 mg·kg^−1^ WW), by [35] in birds in Tuligłowy (43.4 mg·kg^−1^ WW) and by [15] in the liver of mallards harvested in Kanibarazan in Iran (59.63 mg·kg^−1^ DW). Kalisińska et al. [23], in mallards harvested in the Słońsk Reserve, showed a higher average level of this element in juveniles (48.44 mg·kg^−1^ WW) than in adults (42.66 mg·kg^−1^ WW). Dżugan et al. [37], in common pheasants harvested in the vicinity of Szczecin, and Majewska et al. [16], in chicken broilers and turkeys, showed a lower average value for this trait (20.008 mg·kg^−1^ WW, 32.59 mg·kg^−1^ WW, and 33.89 mg·kg^−1^ WW, respectively). Plessl et al. [17] showed a higher average zinc concentration in the liver of mallards harvested in eastern Austria (38.2 mg·kg^−1^ WW). According to [42], the average zinc content in the liver of Ross 308 chicken broilers ranged from 38.7 to 40.9 mg·kg^−1^ WW. Binkowski et al. [14] found that the average zinc content in the liver of mallard ducks obtained in Zatory (40 km from Krakow) was 98.4883 mg·kg^−1^ DW.

The average copper concentration in the breast muscle was significantly higher in mallard ducks shot in the Leszno hunting district than in birds from the Siedlce hunting district. The same relationship was observed in the liver. Mallard ducks harvested in the Leszno hunting district had significantly higher average copper concentrations in the liver than birds from the Siedlce hunting district. This may have been due to emissions of particulate pollution from the Legnica-Głogów Copper District, located in the Lower Silesian Voivodeship. According to [12], toxic elements introduced to the air can be carried over large distances, up to even 1000 km. Similar average copper levels were observed in the breast muscle of male and female mallards. Szymczyk and Zalewski [34] reported a higher average copper level in the breast muscle of mallards harvested in Silesia (6.698 mg·kg^−1^ WW) than was shown in the mallards obtained in the Siedlce hunting district. Alipour et al. [15] and Plessl et al. [17] also reported higher levels of this element in the breast muscle of mallards (20.6 mg·kg^−1^ DW, 5.28 mg·kg^−1^ WW, respectively) than was noted in this species in the two hunting districts in our study. Kokoszyński and Bernacki [13] reported higher average copper concentrations than in the present study in the breast muscle of Pekin ducks, from 23.63 g·kg^−1^ DW for strain P11 to 25.86 g·kg^−1^ DW for strain P22. Kokoszyński et al. [38], in the breast muscles of AF51 and Star 53 H.Y. Pekin ducks, showed average copper levels of 12.5 mg·kg^−1^ DW and 14.4 mg·kg^−1^ DW, respectively. The present study showed a similar copper in the leg muscle of mallard ducks shot in the Siedlce and Leszno hunting areas. The average content of this element in the leg muscles was significantly higher in male mallards than in females. Kokoszyński and Bernacki [13] showed higher average copper levels than in the present study in the leg muscle of Pekin ducks, from 9.38 g·kg^−1^ DW for strain P11 to 9.62 g·kg^−1^ DW for strain P22. Kokoszyński et al. [38] reported higher average copper concentrations in the leg muscles of female SM3 Heavy and AF51 Pekin ducks than was observed in these muscles in female mallards. This may have been due to the fact that utilization of copper from natural feeds amounts to about 20%; for this reason, this micronutrient is added to poultry diets. Bielecka et al. [43] report that on average from 4 mg/kg to 8 mg/kg of copper is introduced to poultry diets in feedstuffs. Among all the tissues analysed in mallards, the highest average copper content was recorded in the liver. This is confirmed by previous research by 27 and by [35]. No statistical differences were shown in the average copper content in the liver between male and female mallards, which is in agreement with findings reported by [17] for birds of this species harvested in autumn in eastern Austria (37 mg·kg^−1^ WW). In the present study, in the Siedlce and Leszno hunting districts, the average copper content in the liver of mallards was higher than that reported by [34] in mallards harvested in Silesia (8.888 mg·kg^−1^ WW) and Warmia-Masuria (7.083 mg·kg^−1^ WW) and by [34] in birds harvested in the vicinity of Lubartów (42.38 mg·kg^−1^ DW). Plessl et al. [17] also observed higher copper concentrations in the liver of mallards harvested in eastern Austria than those obtained for birds harvested in the Siedlce hunting district. According to the authors, the average copper concentration in the liver of mallards was seven times that recorded in the breast muscle. Zalewski et al. [36] in mallards harvested in Warmia-Masuria, and Bojar and Bojar [35], in all study groups of mallards harvested in the Lublin region, showed lower average copper levels in the liver (6.06 mg·kg^−1^ WW and from 3.95 mg·kg^−1^ WW to 13.2 mg·kg^−1^ WW, respectively), depending on where the birds were harvested, than was observed in mallards from the two hunting districts in the present study. The present study showed a higher average copper level in the liver of mallards harvested in the Leszno hunting district than that reported by [15] in mallards harvested in Kanibarazan in Iran. Kalisińska et al. [23] showed a higher average copper level (20.05 mg·kg^−1^ WW) in the liver of mallard ducks shot in the Słońsk Reserve than that recorded in the present study in mallards harvested in the two study areas. According to the authors, such a high average copper concentration in the liver may be explained by the periodic flooding of the Słońsk Reserve by water from the Oder River, which carried pollutants from mines of this element. A study by [44] showed that the Miasteczko Śląskie zinc smelter, located in the Silesian Voivodeship, is more polluted than the Turoszów Coal Basin and the Masurian Lake District, used as a reference. Most kidney and liver samples taken from game animals for analysis exceeded the maximum permissible levels of toxic metals for slaughtered animals. Majewska et al. [16], in the liver of ostriches, turkeys and chicken broilers, reported an average copper level (6.03 mg·kg^−1^ WW, 5.92 mg·kg^−1^ WW and 4.45 mg·kg^−1^ WW, respectively) seven times lower than that observed in mallards harvested in the Leszno hunting district in the present study. Kim and Oh [45], on the other hand, noted an average copper level in the liver of white-fronted geese (86.7 mg·kg^−1^ DW) that was twice that noted in mallards (42.8 mg·kg^−1^ DW).

## 5. Conclusions

It was concluded from the study that the Leszno hunting area is more polluted than the Siedlce hunting area. Among the tissues tested from mallards from both hunting districts, the highest levels of zinc and copper were noted in the liver. In this organ, birds shot in the Leszno hunting district were characterized by significantly higher contents of zinc and copper than individuals caught in the Siedlce hunting district. The significantly higher average zinc and copper levels in the liver of mallard ducks harvested in this hunting district may have been influenced by fertilization of crop fields in this area with fertilizers containing these elements and by extraction of ores containing zinc and copper minerals in the Legnica-Głogów Copper District, located about 80 km away. Taking into account the daily intake of zinc and copper in European countries and the percentage of absorption of the tested elements in the diet, it can be assumed that the content of the elements in the breast and leg muscles does not pose a threat to humans, especially since the consumption of mallard duck meat is sporadic. Only the high content of zinc and copper in the liver of mallard ducks in both hunting districts examined, with a significant degree of absorption of these elements, may concern future consumers of this food. This is an important problem with respect to the safety of consumers of game meat. The average level of zinc and copper in the tissues was higher in male mallards than in females, but the differences were statistically significant only in the case of the average copper content in the leg muscle.

## Figures and Tables

**Table 1 animals-14-01176-t001:** Average zinc content (mg·kg^−1^ WW) in selected tissues and organs of mallard ducks (*Anas platyrhynchos* L.) depending on sex and hunting district.

Tissues and Organs	Basic Statistics	Sex		LSD_0.05_	Hunting District		LSD_0.05_
		Females (*n* = 14)	Males (*n* = 14)		Siedlce (*n* = 12)	Leszno (*n* = 16)	
Breast muscle	$\bar{x}$	9.736 a	11.237 a	n.s.	9.728 a	11.055 a	n.s.
	min.–max.	5.994–17.934	5.404–22.387		5.404–22.387	7.047–17.934	
	s	2.679	4.117		4.168	2.883	
	V%	27.52	36.64		42.85	26.08	
Leg muscle	$\bar{x}$	24.356 a	28.886 a	n.s.	21.309 a	30.605 b	5.983
	min.–max.	15.736–42.041	9.124–44.235		9.124–29.863	15.736–44.235	
	s	7.743	9.438		5.249	9.038	
	V%	31.79	32.67		24.63	29.53	
Liver	$\bar{x}$	34.772 a	35.977 a	n.s.	30.455 a	39.064 b	6.863
	min.–max.	21.334–63.6011	23.774–55.1423		21.334–40.089	23.774–63.601	
	s	10.8122	7.561		6.016	9.698	
	V%	31.09	21.02		19.75	24.82	

Explained: $\bar{x}$—arithmetic mean, x_min._–x_max._—range of variation, s—standard deviation, V%—coefficient of variation, LSD_0.05_—Least Significant Difference (*p* ≤ 0.05), n.s.—not significant, a, b—means with the same letter do not differ significantly (*p* ≤ 0.05).

**Table 2 animals-14-01176-t002:** Average zinc content (mg·kg^−1^ WW) in selected tissues and organs of mallard ducks (*Anas platyrhychos* L.) for the sexes in hunting districts.

Tissues and Organs	Basic Statistics	Hunting District				LSD_0.05_
		Siedlce		Leszno		
		Females (*n* = 6)	Males (*n* = 6)	Females (*n* = 8)	Males (*n* = 8)	
Breast muscle	$\bar{x}$	8.530 a	10.927 a	10.640 a	11.470 a	n.s.
	min.–max.	5.994–10.375	5.404–22.387	7.047–17.934	7.959–14.743	
	s	1.295	5.495	3.065	2.624	
	V%	15.18	50.29	28.81	22.87	
Leg muscle	$\bar{x}$	21.420 a	21.198 a	26.58 a	34.653 a	n.s.
	min.–max.	16.225–29.573	9.124–29.863	15.736–42.041	23.247–44.235	
	s	4.037	6.227	9.021	7.016	
	V%	18.85	29.38	33.97	20.25	
Liver	$\bar{x}$	27.473 a	33.436 a	40.246 a	37.882 a	n.s.
	min.–max.	21.334–37.573	27.471–40.089	31.288–63.601	23.774–55.142	
	s	5.743	4.648	10.484	8.682	
	V%	20.91	13.90	26.05	22.92	

Explained: $\bar{x}$—arithmetic mean, x_min._–x_max._—range of variation, s—standard deviation, V%—coefficient of variation, LSD_0.05_—Least Significant Difference (*p* ≤ 0.05), n.s.—not significant, a, b—means with the same letter do not differ significantly (*p* ≤ 0.05).

**Table 3 animals-14-01176-t003:** Average copper content (mg·kg^−1^ WW) in selected tissues and organs of mallard ducks (*Anas platyrhynchos* L.) depending on sex and hunting district.

Tissues and Organs	Basic Statistics	Sex		LSD_0.05_	Hunting District		LSD_0.05_
		Females (*n* = 14)	Males (*n* = 14)		Siedlce (*n* = 12)	Leszno (*n* = 16)	
Breast muscle	$\bar{x}$	4.537 a	4.706 a	n.s.	3.738 a	5.285 b	1.080
	min.–max.	3.132–6.832	1.579–8.233		1.579–5.341	3.172–8.233	
	s	1.062	1.862		0.985	1.510	
	V%	23.40	39.56		26.34	28.57	
Leg muscle	$\bar{x}$	2.074 a	2.637 b	0.389	2.361 a	2.351 a	n.s.
	min.–max.	1.480–2.785	1.703–3.712		1.480–3.712	1.723–3.244	
	s	0.348	0.569		0.693	0.408	
	V%	16.76	21.57		29.37	17.36	
Liver	$\bar{x}$	31.402 a	31.886 a	n.s.	17.904 a	41.950 b	15.572
	min.–max.	5.114–92.715	5.750–80.360		5.114–35.774	7.365–92.715	
	s	22.584	21.214		10.622	22.533	
	V%	71.92	66.53		59.33	53.72	

Explained: $\bar{x}$—arithmetic mean, x_min._–x_max._—range of variation, s—standard deviation, V%—coefficient of variation, LSD_0.05_—Least Significant Difference (*p* ≤ 0.05), n.s.—not significant, a, b—means with the same letter do not differ significantly (*p* ≤ 0.05).

**Table 4 animals-14-01176-t004:** Average copper content (mg·kg^−1^ WW) in selected tissues and organs of mallard ducks (*Anas platyrhynchos* L.) for sexes in the hunting districts.

Tissues and Organs	Basic Statistics	Hunting District				LSD_0.05_
		Siedlce		Leszno		
		Females (n = 6)	Males (*n* = 6)	Females (*n* = 8)	Males (*n* = 8)	
Breast muscle	$\bar{x}$	4.034 a	3.441 a	4.914 a	5.655 a	n.s.
	min.–max.	3.132–4.831	1.579–5.341	3.172–6.832	3.211–8.233	
	s	0.590	1.190	1.174	1.704	
	V%	14.62	34.57	23.90	30.137	
Leg muscle	$\bar{x}$	1.962 a	2.760 a	2.158 a	2.544 a	n.s.
	min.–max.	1.480–2.341	1.703–3.712	1.723–2.785	2.136–3.244	
	s	0.311	0.739	0.350	0.369	
	V%	15.84	26.78	16.22	14.51	
Liver	$\bar{x}$	14.911 a	20.897 a	43.770 a	40.129 a	n.s.
	min.–max.	5.114–31.696	5.750–35.774	22.114–92.715	7.365–80.360	
	s	9.117	11.164	21.755	23.143	
	V%	61.14	53.43	49.70	57.67	

Explained: $\bar{x}$—arithmetic mean, x_min._–x_max._—range of variation, s—standard deviation, V%—coefficient of variation, LSD_0.05_—Least Significant Difference (*p* ≤ 0.05), n.s.—not significant, a, b—means with the same letter do not differ significantly (*p* ≤ 0.05).

## Data Availability

No new data were created or analysed in this study. Data sharing is not applicable to this article.

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
