# Peer review of "Analysis of Zinc and Copper Content in Selected Tissues and Organs of Wild Mallard Ducks (Anas platyrhynchos L.) in Poland"

_animals, 2024, doi:10.3390/ani14081176_

Round 1
Reviewer 1 Report
Comments and Suggestions for Authors
Dear Authors,
Your paper is interesting and appropriate for publication in the sppecial issue entitled "Environmental and mamagements impacts on animal products", but in my opinion it requires major rewording especially in the context of sources of literwature that were used. Detailed comments are included in attached file.
Your sincerely

Author Response
We are submitting the article: “ANALYSIS OF ZINC AND COPPER CONTENT IN SELECTED TISSUES AND ORGANS OF WILD MALLARD DUCKS (ANAS PLATYRHYNCHOS L.) IN POLAND” autors: Elżbieta Bombik , Antoni Bombik, Katarzyna Pietrzkiewicz, corrected according to the reviewers' instructions and responses to the reviews. We would like to thank for valuable tips that helped us improve our work.
Review 1
Jamroz (2013) and Wojtasik et al. (2017) were removed from the references.
The description of the tables has been corrected.
There are no significant statistical differences in Table 2.
Sincerely
Elżbieta Bombik
University in Siedlce,
14 Prusa Street,
08-110 Siedlce, Poland
Reviewer 2 Report
Comments and Suggestions for Authors
I am not be able to review this manuscript, mainly by three reasons:
1) There are many discordances between references in the text with the list of references (See references #3, #15, #28, #32, #38, #40, #42, and others). That is why I am not sure if the authors cited in the text really have correspondence with the references
2) It is not possible to acced to some references (See references #15, #18, #28, #32, #38, #40 and others).
3) There are in the manuscript too much references in Polish about general and fundamental knowledge of the essential minerals (See references #6, #7, #9, #11, #12, #13, #23, #45). I am sure that there are many references in English to justify these general knowledge. I am not able to review this manuscript with these references in Polish.
In my opinion, authors should improve these aspects before submitting the manuscript to review again.
Comments on the Quality of English Language
Fine
Author Response
We are submitting the article: “ANALYSIS OF ZINC AND COPPER CONTENT IN SELECTED TISSUES AND ORGANS OF WILD MALLARD DUCKS (ANAS PLATYRHYNCHOS L.) IN POLAND” autors: Elżbieta Bombik , Antoni Bombik, Katarzyna Pietrzkiewicz, corrected according to the reviewers' instructions and responses to the reviews. We would like to thank for valuable tips that helped us improve our work.
Review 2
All citations in the text of the work have been corrected in accordance with the list of references.
Items 6 and 7 were removed. The remaining items were left as closely related to the research topic.
Sincerely
Elżbieta Bombik
University in Siedlce,
14 Prusa Street,
08-110 Siedlce, Poland
Reviewer 3 Report
Comments and Suggestions for Authors
Manuscript entitled "Analysis of Zinc and Copper Content in Selected Tissues and Organs of Wild Mallard Ducks (Anas Platyrhynchos L.) in Poland" presents the results of the determination of two selected elements in breast, leg muscles and livers of Wild Ducks harvested in Poland. This results are complementing the pool of knowledge concerning monitoring of metal content in the tissues of wild animals. The idea of the study is good. Title of the manuscript is adequate to the its text. The investigations were done on sufficient animal material using widely accepted methods. All presented tables and references are necessary and adequate. The manuscript should be of great interest to the readers. However, the quality of manuscript would benefit with literature overview in the reference for the last 10 years. For a consumer of this meat would be beneficial to know how consumption of this meat complies with Dietary Reference Values set by European Food Safety Agency or other food authority (USA) for measured minerals.
Below you can find my suggestions for consideration.
Specific comments:
Introduction:
1. L 38 Cite the reference for this sentence.
2. L 81 I suggest removing the term "heavy metals" due to the fact that they are trace elements. “Heavy metals” should be correctly written as e.g. “potentially toxic elements” as stated Pourret, O., Bollinger, JC. & Hursthouse, A. Heavy metal: a misused term? Acta Geochim 40, 466–471 (2021). https://doi.org/10.1007/s11631-021-00468-0
3. L 88 “… in the water column”? difficult sentence
4. What is the importance of these wild ducks in the Poland or European market? It is possible to give in Introduction section how is the volume of consumption of wild duck meat.
5. At the end of the Introduction section, the aim of the research should be written.
Materials and methods:
1. Provide detail information on sampling, during what period, at what age the ducks were. It can to be specific in which year and month was investigation. Please complete this information.
2. Specify the limits of detection and wavelength for each element.
3. Please should clarify in what units the trace elements are presented.
4. L 120 Details lacking in item 26 of the reference.
5. L 139 “standard deviation (s)” ? correct “SD”
Results
1. Why calculations for Zn in liver od male and female are not included in the Table 2?
Discussion
1. L332 “The average content of this element in the leg muscles was significantly higher in male mallards than in females”. Why? add possible reason/mechanism behind the results that authors got. Please explain the reason. Please complete the discussion.
2. Does the content of elements in the analyzed tissues pose a risk to humans, exceed acceptable limits of FAO/WHO ? Add reference from the latest literature from 2019-2024
3. L 369 “average level” what element does it concern?
References
Check references for completeness and consistency, e.g.
L 464 Author, title ?
L 472 Avoid repetition of the information.
L 486 What is the title of the article?
Author Response
We are submitting the article: “ANALYSIS OF ZINC AND COPPER CONTENT IN SELECTED TISSUES AND ORGANS OF WILD MALLARD DUCKS (ANAS PLATYRHYNCHOS L.) IN POLAND” autors: Elżbieta Bombik , Antoni Bombik, Katarzyna Pietrzkiewicz, corrected according to the reviewers' instructions and responses to the reviews. We would like to thank for valuable tips that helped us improve our work.
Review 3
Introduction:
- The publication was cited Kabata-Pendias and Pendias (1999).
- The term "heavy metals" has been replaced with "potentially toxic elements".
- The term “ in the water column” was dropped.
- The introduction added what is the importance of wild ducks on the market.
- At the end of the introduction, the purpose of the research is given.
Materials and methods:
- The information in the text of the work has been supplemented.
- Detection limits and wavelengths for each element are given.
- The content of Zn and Cu was determined in mg·kg−1 WW (wet weight).
- Corrected from Bombik et al. (2022).
- The standard deviation is marked with s.
Results
- It is difficult to indicate the reason for the significantly higher content of this element in the leg muscles of males. There is no available literature on this topic.
- The introduction provides information on the intake of zinc and copper in the human diet. There is no data on the permissible Polish and European standards for the content of zinc and copper in products of animal origin.
- L 369 "medium level" applies to copper. Improvements have been made in the text.
References
The references have been revised in accordance with the reviewers' comments.
Sincerely
Elżbieta Bombik
University in Siedlce,
14 Prusa Street,
08-110 Siedlce, Poland
Reviewer 4 Report
Comments and Suggestions for Authors
The paper presented for evaluation is important in terms of the possibility of monitoring of heavy metal pollution in the environment as well as consumers of game meat. However, it needs to be corrected due to some inaccuracies.
The title of the article is too general due to not the whole country (Poland) were subjected to analysis but only 2 chosen regions.
The introduction does not justify the choice of the experimental material properly. Some less important details about zinc and/or copper should be replaced with the possibility of the animal use as bioindicator of environmental pollution. Also the aim of the study should be precisely given.
M&M: 2-3 sentences about chosen hunting districts should be added, why they were chosen, what is a difference between them. Such kind of information may play the role of research hypothesis. Probably paper https://doi.org/10.1038/s41598-023-33649-3 would be helpful in this case.
In the paragraph about statistical analysis information about testing of data normality should be added.
The authors should consider whether it is reasonable to combine birds from 2 hunting districts and to compare males to females (table 1). It would be justified in the case of any productive traits such as body weight, breast muscle proportion etc. but no differentiation should be expected in the case of mineral content. These differences will disappear because of the different origins of the birds. Also due to above the sentence in lines 386-389 seems to be redundant.
If 4 means are compared (tables 2 and 4) and no statistically significant differences were found there is no need to indicate it with letter symbols (same letter for all means). Only significant differences should be marked.
The legend of tables should be unified as well as self-explanatory.
Line 283 – maybe “the Greater Poland Voivodeship” mean, not all own names should be translated, probably it would be better (more understandable) to keep the original name.
References – the most of them are chosen correctly but authors should to avoid books’ quotations (references: 4, 5, 6) as well as some of them are incorrectly formatted (i.e. 27-28).
Some formatting mistakes were found – i.e. comas instead of dot as a decimal point (table 1), hunting district repetition (in the head of table) in tables 3 and 4.
Once the manuscript has been completed and the necessary revisions made, the work will be suitable for publication in “Animals”.
Author Response
We are submitting the article: “ANALYSIS OF ZINC AND COPPER CONTENT IN SELECTED TISSUES AND ORGANS OF WILD MALLARD DUCKS (ANAS PLATYRHYNCHOS L.) IN POLAND” autors: Elżbieta Bombik , Antoni Bombik, Katarzyna Pietrzkiewicz, corrected according to the reviewers' instructions and responses to the reviews. We would like to thank for valuable tips that helped us improve our work.
Review 4
- We suggest leaving the title unchanged. The methodology shows in which areas the research was conducted.
- The introduction of the work has been improved. The introduction presents the purpose of the research and its justification.
- A response to this comment regarding the characteristics of hunting districts is included in the text of the work.
- The data presented in the work were measurable (quantitative), and a normal distribution of these data was assumed.
- The results were developed and summarized in accordance with the adopted mathematical model, therefore we propose to leave the tables and their description in this form.
- We propose leaving the accepted markings (to standardize them).
- The legend to the tables has been unified and corrected.
- Line 283 - in previous works, the "wielkopolskie" region was translated into English, we propose to leave it in this form.
- References - unnecessary references have been removed. All literature was formatted correctly.
- Some formatting errors were found - corrections made to tables 1, 3 and 4.
Sincerely
Elżbieta Bombik
University in Siedlce,
14 Prusa Street,
08-110 Siedlce, Poland
Round 2
Reviewer 1 Report
Comments and Suggestions for Authors
Dear Authors, thank you for taking into account my suggestions in the manuscript,
Best regards,